# The Nrf2 Inhibitor Brusatol Promotes Human Osteosarcoma (MG63) Growth and Blocks EB1089-Induced Differentiation

**DOI:** 10.3390/ijms26199675

**Published:** 2025-10-03

**Authors:** Emily Stephens, Alexander Greenhough, Jason P. Mansell

**Affiliations:** School of Applied Sciences, University of the West of England, Coldharbour Lane, Bristol BS16 1QY, UK; es2366@bath.ac.uk (E.S.); alexander.greenhough@uwe.ac.uk (A.G.)

**Keywords:** osteosarcoma, brusatol, Nrf2, EB1089, hypoxia, growth, differentiation, alkaline phosphatase

## Abstract

Survival rates for those with metastatic osteosarcoma (OS) have not improved over the last four decades. It is imperative that novel approaches to treating and curing OS be sought. We, therefore, turned our attention to Brusatol (Bru), a naturally occurring Nrf2 inhibitor reported to elicit anti-cancer effects in a multitude of tumour models. Importantly there is emerging evidence that Nrf2 is implicated in chemoradiotherapy resistance in OS and that inhibiting Nrf2 may represent a desirable route to treating OS. Surprisingly, using the human OS cell line, MG63, we actually found that Bru promoted cell growth. Compared to control, normoxic cultures, the application of Bru (50 nM) over 3 days led to an increase in cell number by approximately 1.7-fold. A similar outcome occurred for cells under hypoxic conditions, although the extent of cell growth was significantly less at around 1.3-fold. Furthermore, Bru prevented MG63 differentiation in response to co-treatment with the calcitriol analogue, EB1089, and the lipid growth factor, lysophosphatidic acid. The extent of inhibition was profound at approximately 2.8-fold. The application of the Nrf2 activator, dimethyl fumarate, did not rescue these phenotypes. Whilst Bru has shown promise in other cancer models, it would appear, from our findings, that this agent may not be suitable for the treatment of OS.

## 1. Introduction

With an overall annual incidence rate of 3.1 per million, osteosarcoma (OS) is the most common primary bone cancer [1]. The 5-year survival rate continues to be around 20%, an outcome matching that reported back in the 1950s [2]. Those most commonly affected are adolescents with a peak incidence of 4.2 per million, which then declines to 1.7 per million between the ages of 25–59 [1]. Mutations in the *TP53* and *RB1* genes of osteoblasts and their precursors are deemed to be responsible for solid lesions in the long bones such as the tibia and femur [3,4,5]. Chemotherapy is the first-line treatment option for OS, with patients typically receiving a combination of methotrexate, cisplatin, and doxorubicin [6].

Despite concerted efforts to develop new therapies, there has been no improvement in survival rate over the last four decades. Indeed, in a recent review detailing the plethora of alternative approaches to tackling OS, Harris and Hawkins [6] make the very frustrating remark that “this work has produced negligible improvements”. It is clear that there needs to be a continued effort in finding novel approaches to treating and curing OS.

To this end we turned our attention to brusatol (Bru), a naturally occurring quassinoid, initially sourced and characterised from the seeds of the evergreen shrub, *Brucea sumatrana* in 1968 [7]. Species of *Brucea*, known as Ya-Dan-Zi [8], have a long medical history in China, being used to treat a wide variety of conditions owing to their diverse pharmacological activity [9]. Bru is widely recognised for its anti-cancer effects, a property likely attributed to its inhibitory activity of Nuclear Factor (erythroid-derived 2)-Like 2 (Nrf2) [10]. Interestingly, Nrf2 has been reported to be markedly elevated in OS cells and negatively correlates with five-year patient survival rates [11]. It has been suggested, therefore, that Nrf2 might be an OS oncogene, fueling OS progression and chemoradiotherapy resistance [12]. Given the findings that an increase in Nrf2 expression contributes to chemoresistance, albeit for other cancer cell types [13,14], it is tempting that inhibiting Nrf2 may represent a desirable OS treatment option. There are numerous studies reporting on the anti-cancer effects of Bru on a wide range of malignant cell types. For a recent and comprehensive review of Bru in cancer biology, the reader is referred to [9]. Despite the wealth of interest in Bru as an anti-cancer agent, very little is known about its potential for tackling OS, an aggressive mesenchymal malignancy primarily affecting children and adolescents. One likely explanation for the lack of research on Bru in OS could be linked to the impact of OS against other more prevalent cancers, for example, colorectal and breast cancer. We, therefore, sought to ascertain the effect of Bru on OS cell growth using the human OS cell line, MG63. An assessment of these responses under hypoxia was also examined, given that solid tumours, including OS, experience low oxygen tension, which fuels malignant progression [15]. It is also important to note that hypoxia induces chemoresistance; hypoxia inducible factor-1α (HIF-1α) has been shown to enhance the expression of genes that contribute to chemoresistance, for example, members of the ATP-binding cassette transporters [16]. In assessing the potential of any given compound on OS cell viability, it is pertinent to assess efficacy in a hypoxic setting. Given the widely known anti-cancer and pro-differentiating effects of vitamin D [17], we also wanted to examine the potential interaction between Bru and vitamin D signalling in a model of MG63 differentiation. We selected the vitamin D analogue EB1089, a less calcaemic molecule which has been used clinically in the treatment of cancer [18]. Our model of differentiation consists of co-treating MG63 cells with EB1089 and the pleiotropic lipid growth factor, lysophosphatidic acid (LPA). This results in a synergistic increase in the expression of alkaline phosphatase [19], a marker of more differentiated cells [20,21].

## 2. Results

### 2.1. Bru Promotes the Growth of MG63 Cells

For conventional cultures (normoxia) the application of Bru (sourced from Merck, Gillingham, UK) at 50 nM and 100 nM stimulated significant increases in cell number. However, when the concentration was raised to 400 nM, this did not reach statistical significance (Figure 1). This was surprising in light of the literature for other cancer cell lines, as we anticipated a reduction in cell growth for MG63s treated with Bru between 50 and 200 nM. We, therefore, examined the effect of Bru from two other suppliers (Cayman Chemical and TargetMol Chemicals Inc., both sourced from Cambridge Bioscience, Cambridge, UK) to ensure that the initial responses observed were bona fide. These independent sources of Bru (50 nM) also stimulated significant increases in cell growth, which was comparable to the compound originally obtained from Merck (Table 1). Under hypoxic conditions all three sources of Bru (50 nM) also stimulated cell growth, although the extent was significantly less compared to companion, normoxic cultures (Figure 2).

### 2.2. Bru Inhibits MG63 Differentiation

As anticipated, the co-treatment of MG63 cells with LPA (10 μM) and EB1089 (10 nM) resulted in their differentiation, as supported by a clear increase in ALP activity (Figure 3). However, when cells were co-stimulated in the presence of Bru (100 nM), there was a marked inhibition of differentiation matching that for LPA alone (Figure 3).

### 2.3. The Influence of the Nrf2 Activator Dimethyl Fumarate (DMF) on MG63 Growth and Differentiation

Having found that inhibition of Nrf2 promoted growth and suppressed MG63 differentiation, we wished to examine whether Nrf2 activation might have the opposite effect. To achieve this, we used DMF, which activates Nrf2 by preventing KEAP-1-mediated negative regulation of Nrf2 [22]. When the MG63 cells were treated with DMF (50 μM), there was also evidence for increased cell growth (Figure 4). Likewise, when DMF was applied to cells co-treated with EB1089 (10 nM) and LPA (10 μM), the differentiation response was also suppressed (Figure 5).

## 3. Discussion

In marked contrast to the advances made in treating many paediatric malignancies, the prognosis for individuals with metastatic OS is bleak, with very little change in cure rates over the last four decades [23]. It is generally agreed that novel treatment options are urgently needed to markedly improve patient survival. A potential target for the treatment of OS is the key antioxidant transcription factor, Nrf2. In a study involving 102 OS and surrounding non-cancerous bone tissue samples, Nrf2 expression was found to be much more abundant in OS lesions. In contrast, Kelch-like ECH-associated protein 1 (Keap1) expression was greater in non-cancerous tissue compared to OS.

This differential expression between Nrf2 and Keap1 was found to be negatively correlated with the five-year survival rate of OS patients [11]. In the absence of cell stressors, Nrf2 and Keap1 are coupled together in the cytoplasm, but during periods of oxidative stress, Nrf2 is released and translocates to the nucleus, where it upregulates the expression of antioxidant response element (ARE)-dependent genes [24]. There is emerging evidence that the Keap1/Nrf2/ARE system has a complex role to play in OS and that inhibitors of Nrf2 could help in the fight against OS [12]. Using the OS cell line, MG63, we examined the influence of the Nrf2 inhibitor, Bru [10], on proliferation, initially under conventional culture conditions. We found that using Bru at 50 and 100 nM led to a significant increase in cell growth.

During the course of our investigations, we became aware of only one peer-reviewed study examining the effect of Bru on human OS cells. Using 143B and U2OS cell lines, Yuan and colleagues found that Bru, dose-dependently (10–60 nM), inhibited proliferation [25]. Migration and invasion were also reduced when these cells were exposed to Bru between 10 and 30 nM. Furthermore, there was evidence that Bru restricted in vivo xenograft growth when used between 1 and 4 mg/kg. Since these findings conflicted with our initial results, we deemed it necessary to examine the effect of Bru obtained from two other suppliers. Suffice it to say, all three sources of Bru led to similar increases in MG63 growth.

Next, we examined the influence of Bru, from all three suppliers, on MG63 growth under hypoxia, a hallmark of solid tumours and driver of OS malignant progression [15]. It has recently been found that hypoxia increases Nrf2 expression in OS cells via HIF-1α [26]. The application of Bru to hypoxic cells consistently resulted in significant increases in cell growth. However, the extent of growth was less than that observed for companion, normoxic cultures. Our findings for Bru on hypoxic MG63 growth is intriguing and is at odds with what has been reported on the interplay between hypoxia, HIFs and Nrf2; whilst we did not examine Nrf2 activation directly, it would be anticipated that hypoxia would lead to enhanced Nrf2 activation to alleviate hypoxic stress and promote the expression of genes linked to cell proliferation and apoptotic resistance [27]. Inhibiting Nrf2 in a hypoxic setting might be expected to reduce MG63 cell growth below that for hypoxic controls by reducing the expression of ARE-dependent pro-mitogenic genes. However, there are instances where a negative association between Nrf2 and HIFs exist, wherein Nrf2 inhibition can lead to HIF-1α accumulation [27]. If a negative association between HIFs and Nrf2 exists for MG63 cells, this may explain why hypoxic cells grow in response to Bru. Following on from the cell growth studies, we examined the influence of Bru on a model of MG63 differentiation.

Agonists of the vitamin D receptor (VDR), including the calcitriol analogue, EB1089, co-operate with LPA to promote the expression of MG63 ALP [19], a marker of more mature, differentiated cells [20,21]. VDR agonists have the potential, therefore, in treating cancer because of their antiproliferative, pro-differentiating properties [17]. Steering OS cells towards a more mature or differentiated state is predicted to reduce growth, and we recently reported that exposure of MG63 cells to a VDR agonist led to a marked reduction in cell numbers, including hypoxic cultures [28]. As we have reported previously, the co-stimulation of normoxic MG63 cells with EB1089 and LPA led to clear, synergistic increases in ALP activity.

When cells were co-treated in a hypoxic setting, the extent of differentiation was markedly reduced, something that we reported recently when working with a different VDR agonist based on lithocholic acid [28]. When Bru was co-administered along with LPA and EB1089 to hypoxic cells, there was no change to the differentiation response. Interestingly, Bru actually blunted MG63 differentiation under conventional, normoxic conditions, implying that Nrf2 plays some role in the expression of ALP.

In light of these findings, we wondered whether activating Nrf2 might prevent the negative impact of hypoxia on MG63 maturation. To this end we pivoted to using dimethyl fumarate (DMF). Originally used as a desiccant and fungicide in sofa transportation, DMF is a potent Nrf2 activator and FDA-approved drug for the treatment of psoriasis and multiple sclerosis [24,29]. Of particular relevance to this study are the reports that DMF can bolster the pro-differentiating effects of vitamin D derivatives, albeit for acute myeloid leukaemia cells [30,31]. It was tempting to speculate that DMF might enhance the efficacy of EB1089 and therefore promote hypoxic MG63 cell differentiation. As with Bru, DMF promoted MG63 growth, with greater cell numbers achieved for normoxic cultures. One possible explanation that may account for MG63 growth in response to DMF could be linked to p53 status; MG63 cells lack the tumour suppressor p53 [32]. In a murine model lacking p53, Nrf2 activation triggered the development of oral squamous cell carcinoma along with lesions to the oesophagus, oropharynx and stomach [33]. Thus, in the context of p53 loss, Nrf2 activation has an oncogenic function which may apply to p53 null OS cells. Similarly, DMF blocked the ability of EB1089 and LPA to generate mature cells, as evidenced by a profound decline in ALP activity. There are conflicting reports on the role of Nrf2 in osteoblast differentiation. During periods of oxidative stress, NRf2 has been reported to inhibit the expression of an essential gene linked to bone matrix mineralisation, i.e., ALP. Conversely, Nrf2 maintains redox homeostasis in response to oxidative stress, thereby protecting cells and supporting differentiation [34]. In this latter scenario, inhibition of Nrf2 would lead to compromised differentiation, which would be associated with reduced ALP activity. To summarise we have found that both an inhibitor and activator of Nrf2 can stimulate MG63 growth and prevent their differentiation in response to EB1089-LPA co-stimulation. It is unclear at present as to how agents with opposing actions on Nrf2 activity essentially result in similar MG63 responses. The reliance on a pharmacological approach to target Nrf2 is a clear limitation of this study. To disentangle the findings observed for Bru and DMF, it would be important to consider future genetic approaches to both silence and stably transfect cells with Nrf2.

## 4. Materials and Methods

### 4.1. General

The details provided below are similar to those reported by us previously [28]. Unless stated otherwise, all reagents were of analytical grade from Merck (Gillingham, Dorset, UK). Stocks of EB1089 (10 μM) were prepared in ethanol and stored at −20 °C. Stocks of lysophosphatidic acid (LPA) were prepared in 1:1 ethanol/tissue culture-grade water to a final concentration of 10 mM and likewise stored at −20 °C. Brusatol (Cambridge Bioscience, Bar Hill, UK) stocks (50 and 100 μM) were prepared in ethanol and stored at −20 °C, as were stocks (50 mM) of dimethyl fumarate (DMF, Cambridge Bioscience, Bar Hill, UK).

### 4.2. Human Osteosarcoma (OS) Cells

MG63 human osteosarcoma cells were cultured in 75 cm^2^ tissue culture flasks (Corning, Appleton Woods, Birmingham, UK) under standard conditions of 37 °C and 5% CO_2_ in a humidified incubator. The cells were expanded to confluence using Dulbecco’s Modified Eagle Medium (DMEM)/F12 (Gibco, catalogue no. 21331-020, Paisley, Scotland), supplemented with 1 mM sodium pyruvate, 4 mM L-glutamine, 100 ng/mL streptomycin, 0.1 U/mL penicillin, and 10% (*v*/*v*) foetal calf serum (Gibco). Additionally, the 500 mL culture medium was supplemented with 5 mL of a 100× non-essential amino acid stock solution.

Upon reaching confluence, the cells were transferred into sterile 24-well plates (CytoOne^®^, Starlab, Milton Keynes, UK), with each well receiving 1 mL of a 10,000 cells/mL suspension, quantified via haemocytometer. After a 3-day incubation period, the medium was replaced with serum-free, phenol red-free DMEM/F12 (SFCM; Gibco, catalogue no. 11039-021) to induce overnight starvation.

Subsequent treatments included EB1089 (10 nM), lysophosphatidic acid (LPA, 10 μM), brusatol (Bru, 50 and 100 nM), dimethyl fumarate (DMF, 50 μM), or combinations thereof. All the treatments were administered in SFCM supplemented with 500 μg/mL fatty acid-free human serum albumin. After 72 h, cell monolayers were harvested for quantification of cell number and total alkaline phosphatase (ALP) activity, serving as indicators of cell growth and differentiation, respectively. Experiments were conducted under both normoxic and hypoxic conditions.

### 4.3. Hypoxia Treatments

MG63 cells were exposed to hypoxic conditions (1% O_2_) within a Don Whitley H45 Hypoxystation, which maintained a controlled environment composed of 94% nitrogen, 5% carbon dioxide, and 1% oxygen at a temperature of 37 °C. Throughout the incubation, relative humidity was kept near 75% to support optimal cell culture conditions.

### 4.4. Cell Number

Cell quantification was performed using a colorimetric assay involving the tetrazolium salt MTS (3-(4,5-dimethylthiazol-2-yl)-5-(3-carboxymethoxyphenyl)-2-(4-sulfophenyl)-2H-tetrazolium, inner salt; Promega, UK) in conjunction with the electron coupling agent phenazine methosulfate (PMS). Both reagents were individually dissolved in pre-warmed (37 °C) SFCM. The working solution was prepared by mixing 1 mL of PMS (1 mg/mL) with 19 mL of MTS (2 mg/mL).

MG63 cells were suspended at 1 × 10^6^ cells/mL in SFCM and serially diluted to generate a range of known concentrations down to 2.5 × 10^4^ cells/mL. For each dilution, 0.5 mL of cell suspension was transferred into microcentrifuge tubes and treated with 0.1 mL of the MTS/PMS mixture. Tubes were incubated for 45 min under standard culture conditions. Following incubation, samples were centrifuged at 900 rpm to pellet the cells, and 0.1 mL of the resulting supernatant was transferred to a 96-well plate. Absorbance was measured at 492 nm using a multi-well plate reader.

A standard curve was generated by plotting absorbance values against cell counts previously determined via haemocytometry, allowing estimation of cell numbers in subsequent experimental samples.

### 4.5. ALP Activity

Alkaline phosphatase (ALP) activity was quantified by monitoring the enzymatic conversion of p-nitrophenylphosphate (p-NPP) to p-nitrophenol (p-NP) under alkaline conditions. Following completion of the cell viability assay, residual MTS/PMS reagent was discarded, and cell monolayers were rinsed twice with 0.2 mL of SFCM per well, each wash lasting 5 min, to eliminate remaining formazan.

After washing, cells were lysed by adding 0.1 mL per well of a lysis buffer containing 7 mM sodium carbonate, 3 mM sodium bicarbonate (pH 10.3), and 0.1% (*v*/*v*) Triton X-100. Two minutes later, 0.2 mL of substrate solution—15 mM p-NPP (di-Tris salt, Merck, Gillingham, Dorset, UK) in 70 mM sodium carbonate, 30 mM sodium bicarbonate (pH 10.3) with 1 mM MgCl_2_—was added to each well. The plates were incubated for 1 h under standard culture conditions.

Post-incubation, 0.1 mL aliquots of the reaction mixture were transferred to a 96-well plate, and absorbance was measured at 405 nm. A calibration curve was generated using a dilution series of p-NP (50–400 μM) prepared in the same buffer, enabling quantification of p-NP generation.

### 4.6. Statistical Analysis

Unless otherwise stated, the experiments described above were performed at least three times, using a different passage number of cells, each on different days. Data were deemed to be statistically significant when *p* < 0.05. All analyses were processed using GraphPad Prism 10.4. A one-way ANOVA was generated for Figure 1, Figure 2 and Figure 4, whereas a two-way ANOVA was generated for Figure 3 and Figure 5. For the findings presented in Table 1, the data obtained between the vehicle control and the different suppliers of Bru were subjected to an unpaired, two-tailed Student’s *t*-test.

## 5. Conclusions

The naturally occurring quassinoid, Bru, continues to gather a lot of interest as a potential anti-cancer agent. It is likely that the anti-cancer effect of Bru is linked to its inhibition of Nrf2, a key transcription factor of cell protection, detoxification and metabolic support. In the context of OS, evidence has emerged that Nrf2 could have a negative correlation with the 5-year survival rate. Therefore, inhibiting Nrf2 might have some benefit in treating OS. To address this possibility, we treated a human OS cell line, MG63, with nanomolar (50–400 nM) concentrations of Bru. In marked contrast to other cancer cell lines, we found that Bru actually promoted the growth of MG63s. We also found that Bru could markedly antagonise the pro-differentiating effect of the vitamin D analogue, EB1089. We offer a cautionary note on the applicability of Bru for OS.

## Figures and Tables

**Figure 1 ijms-26-09675-f001:**
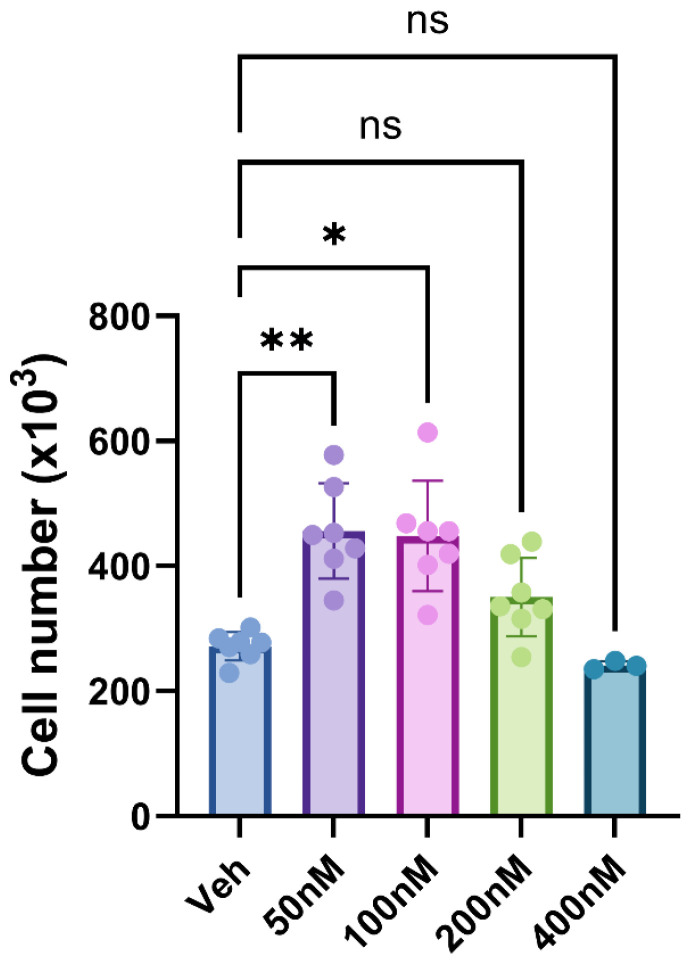
**Brusatol (Bru) promotes osteosarcoma cell growth—**MG63s, seeded into 24-well plates, were treated with Bru for 72 h prior to an assessment of cellularity using the MTS-PMS assay. At 50 and 100 nM, Bru significantly increased cell growth compared to the vehicle control (* *p* = 0.01, ** *p* < 0.005, ns = no significant difference). With the exception of the 400 nM treatment group (means from three independent experiments, 18 replicant samples), the data are the mean values from seven independent experiments (42 replicants) plus the standard deviation.

**Figure 2 ijms-26-09675-f002:**
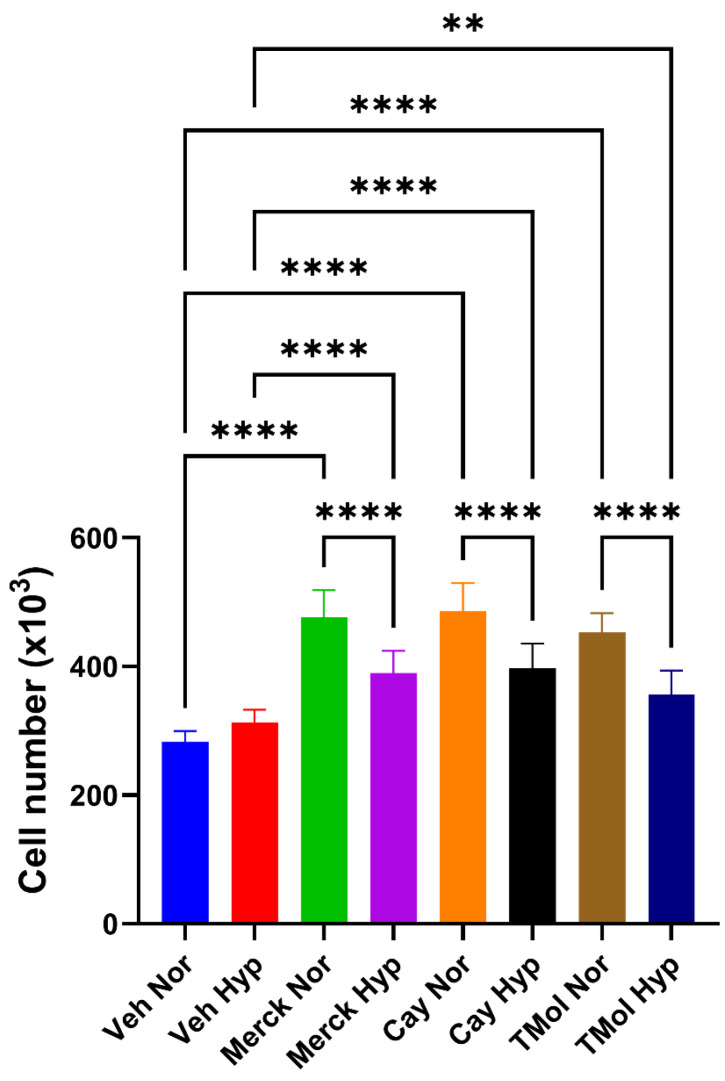
**The influence of hypoxia on Bru-stimulated cell growth.** MG63s, seeded into 24-well plates, were treated with Bru (50 nM) from each of the three different suppliers and left to grow for 72 h under normoxic conditions and compared to parallel cultures maintained in a hypoxia station set at 1% O_2_. As anticipated, Bru promoted significant cell growth compared to vehicle controls (Veh) under normoxic conditions. A similar outcome was observed for hypoxic cells, although the extent of cell growth was significantly less compared to normoxia. The data are the mean values from three pooled experiments (18 replicants) plus the standard deviation. ** *p* < 0.005, **** *p* < 0.0001. Cay—Cayman Chemical. TMol—TargetMol Chemicals Inc.

**Figure 3 ijms-26-09675-f003:**
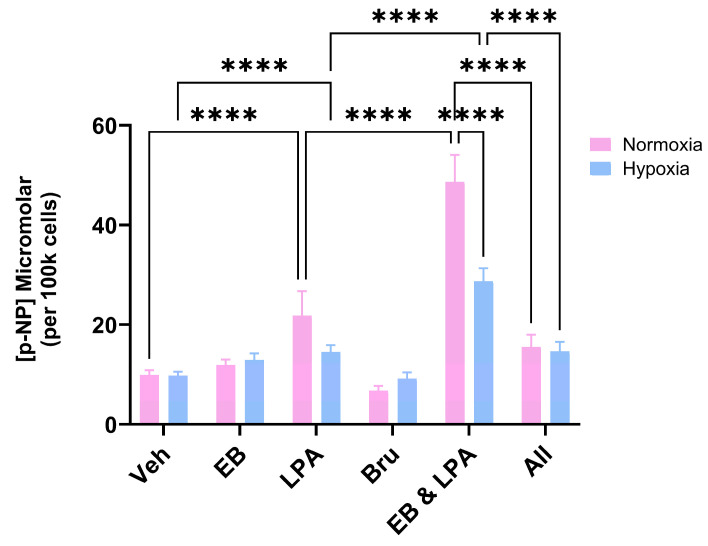
**Brusatol prevents OS differentiation.** MG63 cells were treated with EB1089 (10 nM), LPA (10 μM), Bru (100 nM) or combinations of these for 3 days prior to an assessment of differentiation. The generation of p-nitrophenol (p-NP) from p-nitrophenyl phosphate reflects ALP activity, a marker of the mature osteoblast phenotype. As expected, the co-stimulation of normoxic MG63s with EB1089 and LPA led to a clear increase in ALP activity compared to the agents in isolation. The inclusion of Bru (All) markedly reduced the differentiation response. The data are the mean values from three pooled experiments (18 replicants) plus the standard deviation. **** *p* < 0.0001.

**Figure 4 ijms-26-09675-f004:**
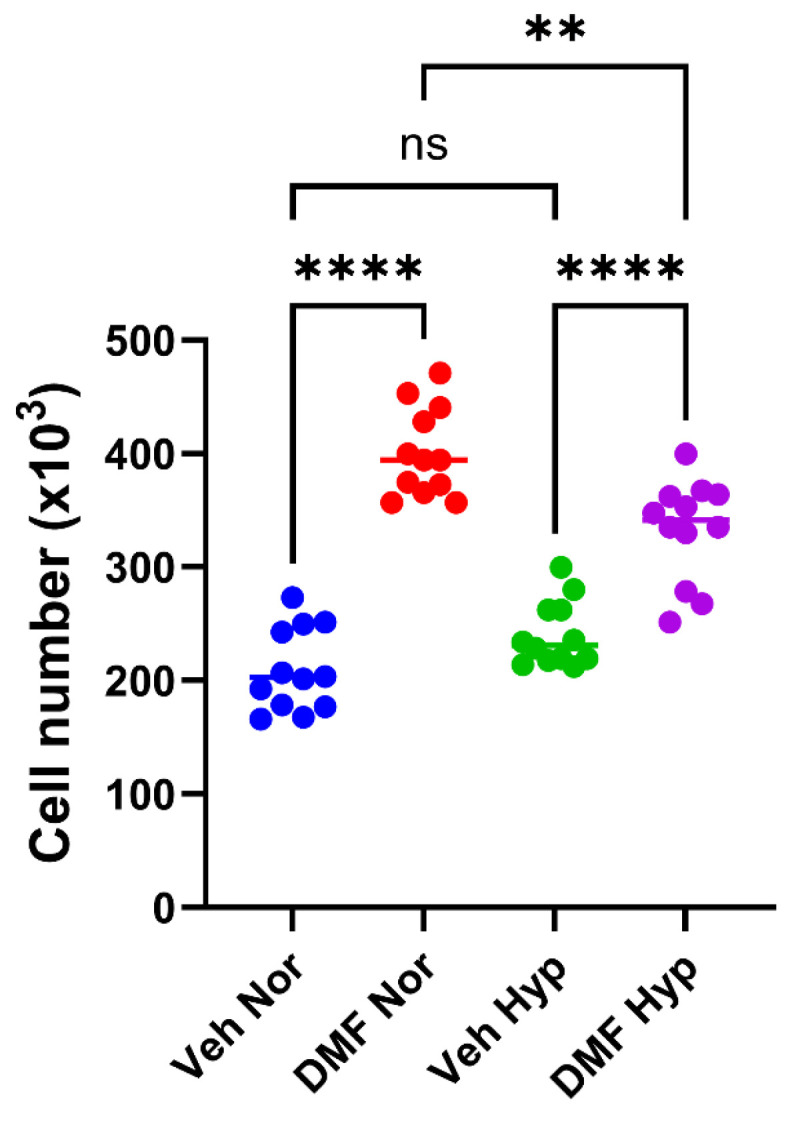
**The Nrf2 activator, dimethyl fumarate (DMF), promotes OS cell growth**. MG63s were treated with DMF (50 μM) for 72 h under normoxic (Nor) and hypoxic (Hyp) conditions, and cell number was determined using the MTS-PMS assay. The data are the mean cell number (×10^3^) pooled from three independent experiments (12 replicants). ** *p* < 0.005, **** *p* < 0.0001, ns = no significant difference.

**Figure 5 ijms-26-09675-f005:**
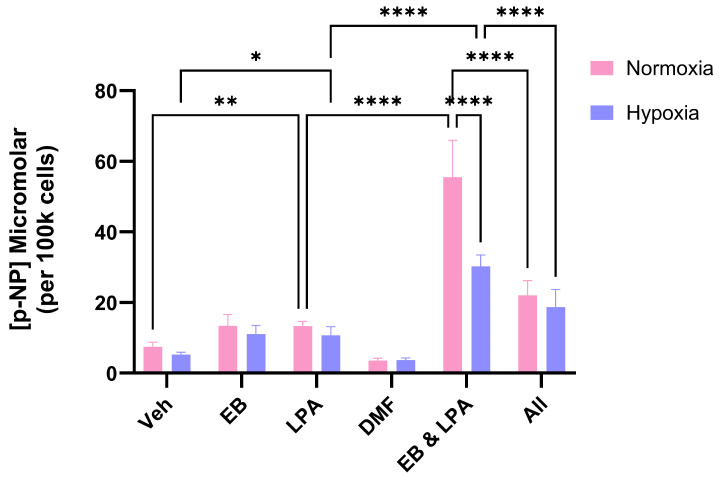
**DMF inhibits OS differentiation.** MG63 cells were treated with EB1089 (10 nM), LPA (10 μM), DMF (50 μM) or combinations of these for 3 days prior to an assessment of differentiation. The generation of p-nitrophenol (p-NP) from p-nitrophenyl phosphate reflects ALP activity, a marker of the mature osteoblast phenotype. As expected, the co-stimulation of normoxic MG63s with EB1089 and LPA led to a clear increase in ALP activity compared to the agents in isolation. The inclusion of DMF (All) markedly reduced the differentiation response. The data are the mean values from two pooled experiments (12 replicants) plus the standard deviation. * *p* < 0.05, ** *p* = 0.006, **** *p* < 0.0001.

**Table 1 ijms-26-09675-t001:** **Brusatol promotes MG63 growth: an assessment from three different sources.** Having found that Bru, originally sourced from Merck, could stimulate MG63 growth, we examined whether other suppliers of Bru could generate a similar outcome. Cells were treated with Bru (50 nM) for 72 h prior to an assessment of cell growth using the MTS-PMS assay. Each of the sources of Bru led to significant increases in cell number compared to the vehicle control (*** *p* < 0.0001). For each experiment, the data are the mean cell number (×10^3^) from six replicates ± SD.

Source of Brusatol	Experiment 1	Experiment 2	Experiment 3
Merck	449.5 ± 14.9 ***	452.8 ± 11.5 ***	526.5 ± 34.2 ***
Cayman Chemical	463.8 ± 15.5 ***	458.4 ± 29.1 ***	536.9 ± 26.7 ***
TargetMol Chemicals Inc	440.0 ± 3.5 ***	429.6 ± 15.9 ***	489.3 ± 16.2 ***
**Vehicle control**	270.6 ± 9.3	277.4 ± 13.7	301.5 ± 5.9

## Data Availability

The datasets generated for this study are available from J.P.M. on reasonable request.

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
