# Peer review of "The Nrf2 Inhibitor Brusatol Promotes Human Osteosarcoma (MG63) Growth and Blocks EB1089-Induced Differentiation"

_ijms, 2025, doi:10.3390/ijms26199675_

Round 1
Reviewer 1 Report
Comments and Suggestions for Authors
Osteosarcoma (OS) remains a challenging cancer to treat. In this context, Stephens et al. investigated the effects of brusatol (Bru), an Nrf2 inhibitor, on human osteosarcoma cells. Their study indicated that Bru promotes cell growth and prevents MG63 cell differentiation in response to co-treatment with EB1089 and lysophosphatidic acid. The use of the Nrf2 activator, dimethyl fumarate, did not modify these outcomes. Although Bru has been effective in other cancer models, their results suggest that it may not be suitable for OS treatment.
This study has several limitations.
1. Introduction: Page 1, Lines 24-26: Why do survival rates for OS remain unchanged?
2. Introduction: Page 1, Lines 32-36: Why have efforts to develop new therapies produced negligible improvement?
3. Introduction: Page 2, Lines 48-53: Discuss the reasons for the lack of research on Brusatol in OS and its potential significance.
4. Results: Page 3, Figure 1: Why does the growth-promoting effect of brusatol reach statistical significance at higher concentrations (200 nM and 400 nM)?
5. Results: Page 3, Table 1: How do you ensure that the observed effects are not due to impurities or variations in brusatol (from different suppliers)?
6. Page 5, Figure 3: How does brusatol prevent MG63 differentiation in response to EB1089 and LPA co-treatment?
7. Results: Page 6, Figure 5: Why does the Nrf2 activator DMF produce effects similar to those of the Nrf2 inhibitor brusatol (Figure 3)? Examination of the possibility of off-target effects or complex feedback mechanisms in Nrf2 signaling.
8. Discussion: Page 7, Lines 194-200: How does hypoxia influence the effects of Brusatol on MG63 cells?
9. Discussion: Page 7, Lines 205-216: Elaborate on the importance of differentiation in osteosarcoma progression and treatment.
10. Discussion: Page 7, Lines 2016-234: What are the potential mechanisms by which both Nrf2 inhibition and activation lead to similar cellular responses in MG63 cells? Please propose and discuss the possible molecular pathways or feedback loops that could explain these unexpected results.
Addressing these concerns, providing a more comprehensive analysis of the results, expanding the discussion within the context of the existing literature, exploring potential mechanisms, and highlighting the implications of these findings for osteosarcoma research may contribute to advancing knowledge in this field.
Author Response
Comment 1. Introduction: Page 1, Lines 24-26: Why do survival rates for OS remain unchanged?
Response 1 – The authors have provided some details regarding this, please see lines 39 to 41 of the revised manuscript. It is important that we highlight the current challenge around OS survival rates. However, a comprehensive breakdown towards explaining the possible reasons for this are beyond the bounds of an introduction for a research article.
Comment 2. Introduction: Page 1, Lines 32-36: Why have efforts to develop new therapies produced negligible improvement?
Response 2 – It is possible that many of the new approaches taken are not particularly well- targeted and/or that OS cells are resistant to selected agents. It would be ill advised of the authors to speculate on what the possible mechanisms might be to account for these “negligible improvements”. Our intention is to provide a clear narrative that new approaches be sought to tackling OS in light of the problems we highlight in the opening paragraph.
Comment 3. Introduction: Page 2, Lines 48-53: Discuss the reasons for the lack of research on Brusatol in OS and its potential significance.
Response 3 – One possible explanation could be linked to the impact of OS against other cancers. In marked contrast to research on colorectal or breast cancer, for example, OS does not share the same level of funding. This is likely attributed to the number of reported cases. We have therefore revised the introduction. This new section, from line 58, now reads: “…adolescents. One likely explanation for the lack of research of Bru in OS could be linked to the impact of OS against other more prevalent cancers, for example colorectal and breast.”
Comment 4. Results: Page 3, Figure 1: Why does the growth-promoting effect of brusatol reach statistical significance at higher concentrations (200 nM and 400 nM)?
Response 4 – From the findings presented it is the lower concentrations of Brusatol (50nM and 100nM) that reach statistical significance. Increasing the concentration to 200nM or 400nM does not.
Comment 5. Results: Page 3, Table 1: How do you ensure that the observed effects are not due to impurities or variations in brusatol (from different suppliers)?
Response 5 – Put quite simply, we can’t ensure that the observed effects are attributed to impurities/variations for the different Brusatol preparations. What is important is that each of the different sources of Brusatol elicit a very similar effect on MG63 cell growth.
Comment 6. Page 5, Figure 3: How does brusatol prevent MG63 differentiation in response to EB1089 and LPA co-treatment?
Response 6 – The authors are currently unclear as to why Brusatol prevents MG63 differentiation to EB1089-LPA co-treatment. Steps to understanding the mechanism will necessitate further, lengthy study.
Comment 7. Results: Page 6, Figure 5: Why does the Nrf2 activator DMF produce effects similar to those of the Nrf2 inhibitor brusatol (Figure 3)? Examination of the possibility of off-target effects or complex feedback mechanisms in Nrf2 signaling.
Response 7 – These similar findings are perplexing and we make reference to this in the final line of the discussion in the original manuscript. Exploring possible off-target effects and/or complex feedback mechanisms will warrant a comprehensive future investigation.
Comment 8. Discussion: Page 7, Lines 194-200: How does hypoxia influence the effects of Brusatol on MG63 cells?
Response 8 – At this stage of our investigations we are unable to offer a clear explanation for these observations. However, we do attempt to link the connection between HIFs and Nrf2 in that section. We expand on this further from lines 206-210 in the revised manuscript.
Comment 9. Discussion: Page 7, Lines 205-216: Elaborate on the importance of differentiation in osteosarcoma progression and treatment.
Response 9 – The authors have included some further information to help clarify the importance of OS differentiation. We have now revised the discussion. This new section, from line 224, now reads: “…properties [17]. Steering OS cells towards a more mature or differentiated state is predicted to reduce growth and we recently reported that exposure of MG63 cells to a VDR agonist led to a marked reduction in cell numbers, including hypoxic cultures [28].”
Comment 10. Discussion: Page 7, Lines 205-216: Elaborate on the importance of differentiation in osteosarcoma progression and treatment.
Response 10 – This is something that has already been raised (comment 7). Without further detailed studies to address these issues it is virtually impossible to give a reliable molecular explanation for the findings presented.
In response to this reviewer the content included in our revised manuscript is highlighted in yellow.
Reviewer 2 Report
Comments and Suggestions for Authors
- Did the different suppliers of Brusatol provide compounds with varying purity or formulations, and could this explain the unexpected growth promotion in MG63 cells?
- The current data includes Brusatol dosages of 50nM, 100nM, 200nM, and 400nM. Would adding data points for lower dosages (between 0-50nM) provide further insights?
- Previous research suggests that inhibiting Nrf2 reduces cell growth. Your findings show the opposite effect. Can you explain the potential reasons for this discrepancy in the discussion?
- Brusatol and DMF are expected to be Nrf2 inhibitors and activators, respectively. Can you suggest potential mechanisms by which these compounds promote MG63 cell growth?
- You note that hypoxia increases Nrf2 expression, yet Brusatol still promotes growth under these conditions. How can this inconsistency be explained?
- You suggest Nrf2 plays a role in ALP expression, given Brusatol's effect on MG63 differentiation. Can you elaborate on this relationship and propose a potential mechanism?
- DMF's unexpected growth promotion and blockage of differentiation warrant further investigation. What are your ideas for future research in this area, particularly regarding its interaction with EB1089?
- What are the limitations of this study that should be considered when interpreting the results?
Author Response
Comment 1. Did the different suppliers of Brusatol provide compounds with varying purity or formulations, and could this explain the unexpected growth promotion in MG63 cells?
Response 1 – This is a very good point. The formulation originally used from Merck had a purity of ≥ 95%. In light of the unexpected growth-promoting action of this preparation on MG63 cells we turned our attention to two other products with a purity of ≥ 98%. As is evident from the findings presented all three products produced a very similar outcome on MG63 growth. Given that the compound was administered in the nanomolar range we are confident that that the findings presented are attributed to Brusatol.
Comment 2. The current data includes Brusatol dosages of 50nM, 100nM, 200nM, and 400nM. Would adding data points for lower dosages (between 0-50nM) provide further insights?
Response 2 – This reviewer raises an important question. Whilst such a study may provide further insights into the growth-promoting efficacy of Brusatol for MG63 cells we steered towards the concentrations selected based on a variety of studies using Brusatol at these concentrations.
Comment 3. Previous research suggests that inhibiting Nrf2 reduces cell growth. Your findings show the opposite effect. Can you explain the potential reasons for this discrepancy in the discussion?
Response 3 – This is a very important point which we agree is at odds with the literature. We are aware of only one study where Brusatol had no clear effect on cell death. The work by Ren and co-workers, which we cite in our manuscript (reference 10) found that the treatment of A549 cells with Brusatol “did not cause any obvious cell death”.
Comment 4. Brusatol and DMF are expected to be Nrf2 inhibitors and activators, respectively. Can you suggest potential mechanisms by which these compounds promote MG63 cell growth?
Response 4 – We offer a possible explanation for how DMF might promote MG63 maturation. The MG63 cells used in our study lack the tumour suppressor p53 (Masuda et al. 1987). It has been reported that oral squamous cell carcinoma occurred following Nrf2 activation in a murine model lacking p53 (Hamad et al. 2024). It is therefore possible that DMF promotes MG63 growth because they lack p53. We have revised the discussion to identify this possibility. The new section, from line 244, now reads: “…achieved for normoxic cultures. One possible explanation that may account for MG63 growth in response to DMF may be linked to p53 status; MG63 cells lack the tumour suppressor p53 [ new ref 32]. In a murine model lacking p53, Nrf2 activation triggered the development of oral squamous cell carcinoma along with lesions to the oesophagus, oropharynx and stomach [new ref 33]. Thus, in the context of p53 loss, Nrf2 activation has an oncogenic function which may apply to p53 null OS cells.
As to Brusatol promoting MG63 growth, this continues to be a conundrum, and we have yet to find a reason for this intriguing observation.
Comment 5. You note that hypoxia increases Nrf2 expression, yet Brusatol still promotes growth under these conditions. How can this inconsistency be explained?
Response 5 – This is a very interesting point and there may be reasons that go towards explaining our paradoxical findings for hypoxic cells. In a recent review by Bae and colleagues (we cite this in our manuscript, reference 27) there are examples of a negative association between Nrf2 and HIFs. Evidence is provided that NRf2 inhibition can promote HIF-1a accumulation which in turn has the potential to support cell growth. We have revised the discussion to identify this possibility. The new section, from line 215, now reads: “…pro-mitogenic genes. However, there are instances where a negative association between Nrf2 and HIFs exist wherein Nrf2 inhibition can lead to HIF-1a accumulation [27]. If a negative association between HIFs and Nrf2 exists for MG63 cells this may explain why hypoxic cells grow in response to Bru.”
Comment 6. You suggest Nrf2 plays a role in ALP expression, given Brusatol's effect on MG63 differentiation. Can you elaborate on this relationship and propose a potential mechanism?
Response 6 – There is evidence that Nrf2 has a role to play in osteoblast function and differentiation. Unsurprisingly there are reports of both positive and negative effects on osteoblasts. In our original manuscript we only provided evidence for a negative effect of Nrf2. We have now revised the discussion to provide balance for how Nrf2 may influence ALP expression and therefore differentiation. The new section, from line 251, now reads: “…ALP activity. There are conflicting reports on the role of Nrf2 on osteoblast differentiation. During periods of oxidative stress NRf2 has been reported to inhibit the expression of an essential gene linked to bone matrix mineralisation, i.e., ALP. Conversely Nrf2 maintains redox homeostasis in response to oxidative stress, thereby protecting cells and supporting differentiation (new ref Han et al. 2022 [34]. In this latter scenario, inhibition of Nrf2 would lead to compromised differentiation which would be associated with reduced ALP activity.”
Comment 7. DMF's unexpected growth promotion and blockage of differentiation warrant further investigation. What are your ideas for future research in this area, particularly regarding its interaction with EB1089?
Response 7 – Based on the interesting works of Danilenko and colleagues (references 30 & 31) we had thought that DMF might enhance EB1089-induced differentiation. In the absence of such an effect it is entirely possible that the findings from the Danilenko lab are peculiar to myeloid cells, i.e., there is context dependency. It is possible however that other VDR agonists could co-operate with DMF owing to their ability to differentially recruit and/or displace VDR co-activators/repressors. To this end we could start by looking at the same vitamin D-like compounds used by the Danilenko lab.
Comment 8. What are the limitations of this study that should be considered when interpreting the results?
Response 8 – The reliance on a pharmacological approach to inhibit/activate Nrf2 is a significant limitation of the study. It would therefore be pertinent to consider a genetic approach to either silence or stably transfect cells with Nrf2. We have revised the final section of the discussion to cover these issues. The new section, from line 260, now reads: “…similar MG63 responses. The reliance on a pharmacological approach to target Nrf2 is a clear limitation of this study. To disentangle the findings observed for Bru and DMF it would be important to consider future genetic approaches to both silence and stably transfect cells with Nrf2.”
In response to this reviewer the content included in our revised manuscript is highlighted in green.
Reviewer 3 Report
Comments and Suggestions for Authors
ijms-3857151-peer-review-v1
- Abstract: More data results should be presented in the abstract.
- The figure caption and main text should be separated.
- The captions in figures and tables should be revised. It is hard to understand these captions.
- Table 1, units ambiguity.
- The study assumes Bru’s effects are mediated via Nrf2 inhibition but lacks direct evidence.
- Alkaline phosphatase (ALP) activity alone is insufficient to conclude differentiation.
- Add a conclusion.
Author Response
Comment 1. Abstract: More data results should be presented in the abstract.
Response 1 – The authors agree. The abstract has now been revised to include more results. We have provided the fold changes for cell number following treatment with Brusatol. We also provide the fold change in differentiation in response to Brusatol.
Comment 2. The figure caption and main text should be separated.
Response 2 – The authors have adhered to the journal instructions in compiling the manuscript within the given template. Al figure captions sit directly underneath all figures and the table.
Comment 3. The captions in figures and tables should be revised. It is hard to understand these captions.
Response 3 – The authors note an error in the formatting of Table 1. The running title has now been moved to the caption. All other captions sit immediately underneath each of the figures.
Comment 4. Table 1, units ambiguity.
Response 4 – To clarify, the units provided in this table are actual cell numbers. At the end of the caption it states: “For each experiment the data are the mean cell number (x103) from six replicates ± SD.”
Comment 5. The study assumes Bru’s effects are mediated via Nrf2 inhibition but lacks direct evidence.
Response 5 – This is a very valid point and the authors acknowledge the limitation of our study in taking a purely pharmacological approach in assessing the influence of Nrf2 on OS cell growth and differentiation. In the discussion we allude to this. The new section, from line 260, now reads: “…similar MG63 responses. The reliance on a pharmacological approach to target Nrf2 is a clear limitation of this study. To disentangle the findings observed for Bru and DMF it would be important to consider future genetic approaches to both silence and stably transfect cells with Nrf2.”
Comment 6. Alkaline phosphatase (ALP) activity alone is insufficient to conclude differentiation.
Response 6 – Whilst the authors acknowledge that other proteins are expressed during osteoblast/OS cell differentiation, monitoring ALP is a very reliable means of assessing whether these cells are maturing. It is generally agreed that this enzyme is a marker of differentiation, and we cite those early works (references 20 & 21) that underpin this claim in our manuscript.
Comment 7. Add a conclusion.
Response – The authors have now included a conclusion to convey the salient findings of our study. This is a new section, section 5, line 339.
In response to this reviewer the content included in our revised manuscript is highlighted in blue.
Round 2
Reviewer 1 Report
Comments and Suggestions for Authors
The authors adequately addressed all concerns. The paper meets the criteria for publication.
Author Response
The authors are pleased that we have adequately addressed the concerns raised by this reviewer and that our manuscript meets the criteria for publication.
Reviewer 3 Report
Comments and Suggestions for Authors
All questions have been addressed in the revised manuscript.
Author Response
The authors are pleased to learn that all questions have been addressed in our revised manuscript.